# Unveiling Resistance and Virulence Mechanisms under Darwinian Positive Selection for Novel Drug Discovery for *Gardnerella vaginalis*

Eduarda Guimarães Sousa [1,*], Andrei Giacchetto Felice [2,*], Fabiana Vieira Dominici [2], Arun Kumar Jaiswal [1], Mariana Letícia Costa Pedrosa [1], Luiza Pereira Reis [1], Lucas Gabriel Rodrigues Gomes [1], Vasco Ariston de Carvalho Azevedo [1,*] and Siomar de Castro Soares [2,*]

[1] Cellular and Molecular Genetics Laboratory, Institute of Biological Sciences, Department of General Biology, Federal University of Minas Gerais, Pampulha Campus, Belo Horizonte 31270-901, MG, Brazil; arunjaiswal1411@gmail.com (A.K.J.); marianaleticia984@gmail.com (M.L.C.P.); luiza19932004@icloud.com (L.P.R.); lucasgabriel388@gmail.com (L.G.R.G.)

[2] Laboratory of Immunology and Bioinformatics, Institute of Biological and Natural Sciences, Department of Microbiology, Immunology and Parasitology, Federal University of Triângulo Mineiro, Educational Center Campus, Uberaba 38025-180, MG, Brazil; fabianadominici88@gmail.com

[*] Correspondence: eduardaguimaraessousa@gmail.com (E.G.S.); andreigf@hotmail.com (A.G.F.); vascoariston@gmail.com (V.A.d.C.A.); siomars@gmail.com (S.d.C.S.)

**Abstract:** *Gardnerella vaginalis* is a Gram-variable bacillus capable of causing bacterial vaginosis, a condition prevalent in reproductive-age women, this bacterium is present in almost 100% of cases and is also considered a gateway to various sexually transmitted infections. This organism exhibits high pathogenicity linked to virulence and resistance genes acquired throughout evolution, showcasing elevated resistance to a broad spectrum of drug classes. This study conducted comparative genomic analyses to identify these genes and correlate their presence with positive Darwinian selection. Additionally, new drug targets were selected through docking and molecular modeling, guided by the heightened antimicrobial resistance exhibited by this microbial species. The available genomes of *G. vaginalis* were analyzed, and the orthologous genes were delineated and positively selected, whereby 29 groups were found. Of these genes, one of great importance was predicted, *Mef(A)*, which is related to resistance to the macrolide group of antibiotics, which are one of the main choices for the treatment of sexually transmitted infections. Additionally, two potential protein candidates were selected as drug targets. These proteins were linked with a natural compound each and are considered good potential drug targets. The analyses in this study contribute to analyzing the evolution of the species and how resistance genes are related to their permanence as a potential pathogen.

**Keywords:** *Gardnerella vaginalis*; positive selection; drug targets





## 1. Introduction

*Gardnerella vaginalis* is a bacillus-shaped bacterium, commonly described as Gram-variable due to its unpredictable response to Gram staining, that is influenced by the age of cultures and the physiological state of the bacterium. A specific analysis of its cell wall revealed a thin layer of peptidoglycan as it aged, along with the absence of lipopolysaccharide and its components, such as heptose and hydroxylated fatty acids. This characteristic has led many authors to characterize it as Gram-positive. This organism lacks flagella or any other means of locomotion, does not produce spores, and lacks a capsule [1].

This pathogen can potentially cause bacterial vaginosis (BV), a common dysbiosis in the lower genital tract of women of reproductive age. This condition is characterized by a reduction in *Lactobacillus* species in the vaginal region, followed by an increase in facultative anaerobic bacteria. BV has been associated with various women's health issues, including an increased risk of pregnancy and childbirth complications, and contracting

sexually transmitted diseases [2]. For example, BV can increase a woman's risk of acquiring *Neisseria gonorrhoeae*, *Chlamydia trachomatis*, and *Trichomonas* [3]. Despite being considered a dysbiosis caused by different bacterial genera, *G. vaginalis* is the species that predominantly causes BV, detected in samples from affected women in up to 95% of cases [1].

This bacillus has a higher pathogenic potential compared to other organisms, and some studies have already identified the potential of this species, which is often under-estimated. This can be explained by its ability to form biofilms [4], which is important toincrease its chances of survival in the human body [5] by increasing its aherence to vaginal epithelial cells. Bacterial biofilms have the following characteristics: hindering the diffusion of drugs to the core; carrying out phenotypic changes, bacterial communication through quorumsensing, and upregulation of genes that enable antimicrobial resistance; the presence of enzymes that hinder the penetration of drugs into the biofilm; and adaptation in the physiological structure [6–8].

This bacterium also possesses prolidase and sialidase activities. These are enzymes that act as toxins to increase their ability to adhere to and destroy human tissue and have sometimes been associated with premature birth or miscarriages [9–11]. There is also vaginolysin (VLY gene), an important virulence factor that acts by lysing erythrocytes and interfering with the immune system of the host [10,12–14].

Studies have shown a resistance to the main drugs used for the treatment of bacterial vaginosis caused by *G. vaginalis*, mainly a resistance to the main line of treatment against the pathogenesis, metronizadol [15,16]. A study classified different strains of *G. vaginalis* in relation to their susceptibility or lack thereof to metronidazole. The results demonstrated that some strains remained susceptible to the drug, which could be attributed to a shared genetic ancestry [6]. The study also identified a resistance phenotype for nitroimidazoles and aminoglycosides [15]. Strains of *G. vaginalis* exhibiting intermediate resistance to kanamycin have been identified, attributed to the presence of a gene belonging to the aminoglycoside phosphotransferase (APH) family. Additionally, resistance mediated by the *tetM* and *tetL* genes, reducing sensitivity to tetracycline, was observed. Furthermore, genes associated with superfamilies of ABC transporters (ATP Binding Cassette) and PBP (penicillin-binding proteins) were found in these strains, strongly linked to drug resistance [6,15].

These resistance genes may be involved in processes of positive Darwinian selection [17], in addition to those related to the host–pathogen relationship, immunity, and virulence [18], as reported in several important pathogenic taxa, *Escherichia coli* [19], *Campylobacter* [20], and *Salmonella* [21]. In general, studies of this nature are concerned with understanding evolutionary and phenotypic characteristics that have been positively selected.

Comparative genomic studies of *G. vaginallis* have shown a certain gene diversity among the species and phylogenetic analysis showing the common ancestry of the organisms, where a correlation was observed between organisms, namely high genomic plasticity, which is crucial for adaptation and islands of resistance that can be transferred horizontally, thus explaining this plasticity [22–24]. In addition, the species may include nine distinct genotypes (GGtype1 to GGtype9), and this may be related to the virulence and resistance potential of this microbial taxon [25] that can be associated with genome-scale positive selection detection (GSPSD).

In this comparative genomics study, we aim to identify the resistance and virulence phenotypes present in the core and which of these genes are related to Darwinian positive selection. In addition, generating virulence and antibiotic resistance requires identifying new drug targets, so molecular modeling and docking analyses with natural compounds will be carried out concurrently.

## 2. Materials and Methods

### 2.1. Genome Information

We used 97 *Gardnerella vaginallis* Refseq and annotated genomes available from the National Center for Biotechnology Information Datasets (NCBI) (https://www.ncbi.nlm.

nih.gov/datasets/) (accessed on 10 August 2023) for the comparative genomics analyses using. "fna", "faa" and "gbk" formats to perform the studies (Table 1).

**Table 1.** Information about the 97 genomes of *G. vaginalis* strains.

| Assembly Accession | Organism Name | Assembly BioSample Accession |
|---|---|---|
| GCF_002861965.1 | *Gardnerella vaginalis UMB0386* | SAMN08193674 |
| GCF_001049785.1 | *Gardnerella vaginalis 3549624* | SAMN03801593 |
| GCF_001278345.1 | *Gardnerella vaginalis 14019_MetR* | SAMN04014465 |
| GCF_001546455.1 | *Gardnerella vaginalis GED7760B* | SAMN03851015 |
| GCF_001546485.1 | *Gardnerella vaginalis PSS_7772B* | SAMN03851016 |
| GCF_001563665.1 | *Gardnerella vaginalis CMW7778B* | SAMN03851013 |
| GCF_001660735.1 | *Gardnerella vaginalis 23-12* | SAMN04625558 |
| GCF_001660755.1 | *Gardnerella vaginalis 18-4* | SAMN04625602 |
| GCF_001913835.1 | *Gardnerella vaginalis ATCC 49145* | SAMN05757759 |
| GCF_002206225.1 | *Gardnerella vaginalis FDAARGOS_296* | SAMN06173309 |
| GCF_002861165.1 | *Gardnerella vaginalis UMB0061* | SAMN08193668 |
| GCF_002861925.1 | *Gardnerella vaginalis UMB0775* | SAMN08193678 |
| GCF_002861945.1 | *Gardnerella vaginalis UMB0770* | SAMN08193677 |
| GCF_002861975.1 | *Gardnerella vaginalis UMB0298* | SAMN08193676 |
| GCF_002862005.1 | *Gardnerella vaginalis UMB0032B* | SAMN08193675 |
| GCF_002862015.1 | *Gardnerella vaginalis UMB0032A* | SAMN08193673 |
| GCF_002862045.1 | *Gardnerella vaginalis UMB0233* | SAMN08193672 |
| GCF_002884835.1 | *Gardnerella vaginalis UMB0768* | SAMN07511408 |
| GCF_002894105.1 | *Gardnerella vaginalis DNF01149* | SAMN05578253 |
| GCF_002896555.1 | *Gardnerella vaginalis KA00225* | SAMN05578087 |
| GCF_003034925.1 | *Gardnerella vaginalis ATCC 49145* | SAMN08644262 |
| GCF_003369875.1 | *Gardnerella vaginalis KA00225* | SAMN03145604 |
| GCF_003369895.1 | *Gardnerella vaginalis N101* | SAMN03145579 |
| GCF_003369935.1 | *Gardnerella vaginalis N153* | SAMN03145603 |
| GCF_003369965.1 | *Gardnerella vaginalis N95* | SAMN03145504 |
| GCF_003397605.1 | *Gardnerella vaginalis UGent 25.49* | SAMN09373179 |
| GCF_003397665.1 | *Gardnerella vaginalis UGent 09.07* | SAMN09373175 |
| GCF_003408745.1 | *Gardnerella vaginalis GH015* | SAMN04446401 |
| GCF_003408775.1 | *Gardnerella vaginalis N160* | SAMN04446403 |
| GCF_003408785.1 | *Gardnerella vaginalis N165* | SAMN04446402 |
| GCF_003408835.1 | *Gardnerella vaginalis N144* | SAMN04446400 |
| GCF_003408845.1 | *Gardnerella vaginalis NR010* | SAMN04446404 |
| GCF_003585655.1 | *Gardnerella vaginalis NR038* | SAMN07490630 |
| GCF_003585755.1 | *Gardnerella vaginalis NR039* | SAMN07490631 |
| GCF_003812765.1 | *Gardnerella vaginalis FDAARGOS_568* | SAMN10163192 |
| GCF_004336715.1 | *Gardnerella vaginalis 14018c* | SAMN11037839 |
| GCF_013315005.1 | *Gardnerella vaginalis UMB0143* | SAMN15064064 |
| GCF_013315025.1 | *Gardnerella vaginalis UMB0736* | SAMN15064063 |
| GCF_013315045.1 | *Gardnerella vaginalis UMB0540* | SAMN15064062 |
| GCF_013315075.1 | *Gardnerella vaginalis UMB0202* | SAMN15064060 |
| GCF_013315085.1 | *Gardnerella vaginalis UMB0358* | SAMN15064061 |
| GCF_013315115.1 | *Gardnerella vaginalis UMB0558* | SAMN15064059 |
| GCF_014857145.1 | *Gardnerella vaginalis 06-12-0010* | SAMN16294983 |
| GCF_023016185.1 | *Gardnerella vaginalis KC2* | SAMN23424279 |
| GCF_023016205.1 | *Gardnerella vaginalis KC1* | SAMN23424278 |
| GCF_023016225.1 | *Gardnerella vaginalis KC4* | SAMN23424281 |
| GCF_023016245.1 | *Gardnerella vaginalis KC3* | SAMN23424280 |
| GCF_023277565.1 | *Gardnerella vaginalis JNFY17* | SAMN21246408 |
| GCF_023277605.1 | *Gardnerella vaginalis JNFY14* | SAMN21246406 |
| GCF_023277625.1 | *Gardnerella vaginalis JNFY13* | SAMN21246405 |
| GCF_023277645.1 | *Gardnerella vaginalis JNFY11* | SAMN21246404 |
| GCF_023277665.1 | *Gardnerella vaginalis JNFY9* | SAMN21246403 |
| GCF_023277685.1 | *Gardnerella vaginalis JNFY4* | SAMN21246402 |
| GCF_023277725.1 | *Gardnerella vaginalis JNFY1* | SAMN21246400 |

**Table 1.** *Cont.*

| Assembly Accession | Organism Name | Assembly BioSample Accession |
|---|---|---|
| GCF_030213965.1 | *Gardnerella vaginalis UMB9230* | SAMN34996711 |
| GCF_030215405.1 | *Gardnerella vaginalis UMB6972* | SAMN34996565 |
| GCF_030216615.1 | *Gardnerella vaginalis UMB6789* | SAMN34996560 |
| GCF_030217865.1 | *Gardnerella vaginalis UMB1190A* | SAMN34996494 |
| GCF_030218185.1 | *Gardnerella vaginalis UMB1019* | SAMN34996474 |
| GCF_030228365.1 | *Gardnerella vaginalis UMB1218B* | SAMN35153957 |
| GCF_030228445.1 | *Gardnerella vaginalis UMB1190B* | SAMN35153955 |
| GCF_030233905.1 | *Gardnerella vaginalis UMB10121* | SAMN35153918 |
| GCF_900105405.1 | *Gardnerella vaginalis DSM 4944* | SAMN04488545 |
| GCF_900637625.1 | *Gardnerella vaginalis NCTC10287* | SAMEA4535760 |
| GCF_000263555.1 | *Gardnerella vaginalis 0288E* | SAMN02393775 |
| GCF_000263495.1 | *Gardnerella vaginalis 1400E* | SAMN02393779 |
| GCF_000263595.1 | *Gardnerella vaginalis 1500E* | SAMN02393780 |
| GCF_000263435.1 | *Gardnerella vaginalis 284V* | SAMN02393773 |
| GCF_000214315.1 | *Gardnerella vaginalis 315-A* | SAMN00138210 |
| GCF_000165635.1 | *Gardnerella vaginalis 41V* | SAMN02472074 |
| GCF_000263475.1 | *Gardnerella vaginalis 55152* | SAMN02393778 |
| GCF_000263655.1 | *Gardnerella vaginalis 6119V5* | SAMN02393784 |
| GCF_000263535.1 | *Gardnerella vaginalis 75712* | SAMN02393774 |
| GCF_000178355.1 | *Gardnerella vaginalis ATCC 14018* | SAMN02471014 |
| GCF_001042655.1 | *Gardnerella vaginalis ATCC 14018* | SAMD00061047 |
| GCF_003397685.1 | *Gardnerella vaginalis ATCC 14018* | SAMN09373172 |
| GCF_004336685.1 | *Gardnerella vaginalis ATCC 14018* | SAMN11037755 |
| GCF_000159155.2 | *Gardnerella vaginalis ATCC 14019* | SAMN00001462 |
| GCF_000213955.1 | *Gardnerella vaginalis HMP9231* | SAMN00100736 |
| GCF_000414705.1 | *Gardnerella vaginalis JCP7275* | SAMN02436832 |
| GCF_000414685.1 | *Gardnerella vaginalis JCP7276* | SAMN02436904 |
| GCF_000414645.1 | *Gardnerella vaginalis JCP7672* | SAMN02436831 |
| GCF_000414525.1 | *Gardnerella vaginalis JCP8108* | SAMN02436830 |
| GCF_000414465.1 | *Gardnerella vaginalis JCP8481A* | SAMN02436910 |
| GCF_000414445.1 | *Gardnerella vaginalis JCP8481B* | SAMN02436829 |
| GCF_000263615.1 | *Gardnerella vaginalis 00703Bmash* | SAMN02393781 |
| GCF_000263515.1 | *Gardnerella vaginalis 00703Cmash* | SAMN02393782 |
| GCF_000263635.1 | *Gardnerella vaginalis 00703Dmash* | SAMN02393783 |
| GCF_000414665.1 | *Gardnerella vaginalis JCP7659* | SAMN02436712 |
| GCF_000414625.1 | *Gardnerella vaginalis JCP7719* | SAMN02436711 |
| GCF_000414605.1 | *Gardnerella vaginalis JCP8017A* | SAMN02436912 |
| GCF_000414585.1 | *Gardnerella vaginalis JCP8017B* | SAMN02436773 |
| GCF_001546445.1 | *Gardnerella vaginalis GED7275B* | SAMN03851014 |
| GCF_002861905.1 | *Gardnerella vaginalis UMB0830* | SAMN08193679 |
| GCF_002861885.1 | *Gardnerella vaginalis UMB0833* | SAMN08193680 |
| GCF_002884775.1 | *Gardnerella vaginalis UMB1686* | SAMN07511412 |
| GCF_000165615.1 | *Gardnerella vaginalis 101* | SAMN02472073 |

## 2.2. Identification of Orthologues

For the prediction of orthologous genes, the Orthofinder v2.5.4 software application was employed (https://github.com/davidemms/OrthoFinder) (accessed on 20 August 2023), performing a clustering calculation based on the Markov Clustering Algorithm (MCL). This algorithm compared all genomes against each other to define clusters with high levels of similarity within the analyzed data [26]. Associating these results with an *in-house* script (ortho_pangenome_splitter.pl) that classifies genes, three sets were obtained: the core genome, which is present in all analyzed lineages and essential for the microorganism's survival; shared, which gathers genes present in two or more lineages but not in all; and singletons, which are specific to only one lineage [27].

### 2.3. Identification of Positively Selected Genes

POsitive selecTION (POTION) v1.2 (https://github.com/g1o/POTION) (accessed on 14 October 2023) is a massively parallel program that identifies positive Darwinian selection in genomic analyses of groups of homologous genes through phylogenetic comparisons of protein-coding genes. It is an open-source end-to-end pipeline for selecting groups related toh non-synonymous substitution, whereby a mutation occurs that will result in a change in amino acid in the proteins that are subject to natural selection, thus demonstrating which phenotypic patterns have evolved at the molecular level [18]. To measure positive selection at the codon level, it is necessary to discriminate $\omega$, the ratio of nonsynonymous to synonymous substitution rates [28]. Orthofinder's orthogroups.tsv output was used as an input for signaling the homologous groups, and an *in-house* script (ortho2mcl.pl) was used to select the count of each orthogroup and how many times it appeared in each of the lineages.

These orthogroups were used for functional analysis and for the identification of which genes and proteins are mainly related to the resistance phenotype of *G. vaginalis*, using the Basic Local Alignment Search Tool (BLASTp) (https://blast.ncbi.nlm.nih.gov/Blast.cgi) (accessed on 14 October 2023).

### 2.4. Subtractive Genomics and Protein Subcellular Localization Prediction

It was necessary to use the BLASTp algorithm to compare the core genome with the human genome, given the classification of orthologous genes previously performed by Orthofinder, to perform subtractive genomics. This analysis was performed to analyze which protein-coding genes are inserted into the genomes and do not have homology with the host, in this case, the human being, to avoid adverse effects. To continue the analysis of possible targets for drugs [29], an *in house* script (core-non-host.pl) was used to align the amino acid sequences (faa) in analysis against the human genome.

An analysis was performed to predict the subcellular localization of the proteins using SurfG+ software v1.2.1 [30] to define which ones would be analyzed for drug targets. Those proteins related to vital metabolic processes and bacterial survival, which are usually cytoplasmic, were selected for drug target analysis [31].

### 2.5. Genomic Resistance and Virulence Analysis

Pan-Resistome Analysis Pipeline (PRAP) v1.0 (https://github.com/syyrjx-hyc/PRAP) (accessed on 28 October 2023) was used to predict Antibiotic resistance genes (ARGs). This platform-independent Python3 tool predict genes related to resistance using whole genomes based on the Comprehensive Antibiotic Resistance Database (CARD) or ResFinder databases; furthermore, the annotations used for the characterization of pan-resistomes can be used to characterize the distribution of ARGs among the input genomes [32]. Only the CARD was used for Pan-resistome analyses.

Pan Virulence and resisTance Analysis (PanViTa) (https://github.com/dlnrodrigues/panvita) (accessed on 29 October 2023) is a tool that not only predicts ARGs and Pan-resistomes like PRAP, but it also analyzes virulence and antibacterial biocide and metal resistance, using the CARD, Antibacterial Biocide and Metal Resistance Genes Database (BACMET), and Virulence Factor Database (VFDB). For the analyses of the *G. vaginalis* genomes, only virulence analyses were carried out using minimum identity to infer presence ("=70") [33].

### 2.6. Selection of Possible Drug Target Candidates

The Essential Gene Database (DEG) was used to assess whether these candidate proteins for drug targets are translated from genes that are essential for the permanence of the studied bacteria and are indispensable for the biological processes of it [34].

Subsequently, the DrugBank platform (www.drugbank.ca) (accessed on 1 November 2023) was employed to predict proteins with significant interactions and druggability. This database offers comprehensive information on drugs, encompassing their drug interac-

tions, bindings, associations, and mechanisms of action. Within this context, we exclusively selected drug targets whose druggability has been assessed as high [35].

The proteins identified by the VFDB with the virulence phenotype were selected as drug targets since they are essential for the survival and spread of the pathogen to the host [36].

### 2.7. Protein Tertiary Structure Prediction

Through the sequence of the proteins, the prediction of their 3D structure was performed using Alphafold v2.3.2 (https://github.com/google-deepmind/alphafold/tree/main) (accessed on 10 November 2023) [37]. This AI system developed by DeepMind predicts the secondary and later the tertiary structure of the protein in a reliable and reproducible way, where an accurate structure prediction is represented according to multiple sets of sequence alignments (MSAs) and procedures based on evolutionary, geometric, and physical constraints, where all the 3D coordinates of all the protein atoms are predicted from the aligned primary amino acid sequence, according to their distance and interactions compared with the PDB.

The network of this software application comprises two main steps: the first step is the processing of an Multiple sequence alignment (MSA) comparing the number of sequences with the number of residues, using a neural network block called Evoformer to make an N seq × N res matrix (N seq, number of sequences; N res, number of residues); the spatial and evolutionary relationships will be defined. Afterward, it is evaluated according to the rotation and translation of the 3D structure. The model with the highest predicted Local Distance Difference Test (LDDT) score (pLDDT) will be selected, which measures the local distance differences in all the atoms in a model [38,39].

To improve the structure of the selected proteins, GalaxyRefine [40] was used, which refines the side chains by molecular dynamics simulation to improve the quality of the structure. The Ramachandran Diagram predicted using the PROCHECK v.3.5 software confirmed the protein structure by evaluating the stereochemical quality of protein structures [41].

### 2.8. Molecular Modeling Analysis

The Autodock tool (ADT) of the MGLTool package v1.5.7 [42] was used for the 3D structure analysis of the final drug target candidates, where a grid box for each target was constructed to cover the region of the protein active site predicted by the DoGSiteScorer software v2.0 application of the Protein plus server [43,44]. In parallel, 5008 natural compounds were selected as ligands from the ZINC database [45]. The Natural Compound ligand library with 5008 molecules was prepared according to the criteria stated by Lipinski's rule, such as the following: hydrogen bond donors not greater than 5, hydrogen bond acceptors not greater than 10, molecular weight not greater than 500 Da, and octanol/water partition coefficient (log P) not greater than 5 for their drug-like properties. The compound molecules were prepared according to the above-mentioned rule and downloaded from the ZINC database in 2017, and the library was first used in our published work by Alissa de Sarom et al., in 2018 [28], against *Haemophilus ducreyi*. Af. The AutoDock Vina software v1.2.5 application was used to perform the modeling analysis itself [46]. The top 10 molecules were extracted for each target sequentially through an *in-house* Python script and analyzed according to their binding affinity and hydrogen bridges. For visualization of this binding and extraction of the 3D image of the target, the Chimera software v1.17.1 application was used [47].

BIOVIA Discovery Studio v21.1.0.0 [48] was used to create 2D interaction photos of the complex between the select proteins and their respective ligands predicted by Docking in Autodock toll.

### 3. Results

All the steps that were carried out are included in a flowchart. A total of 97 *Gardnerella vaginalis* genomes were compared using the methodology inserted in Figure 1, which

summarizes the proteins selected for molecular docking and the orthogroups selected for positive selection analysis.

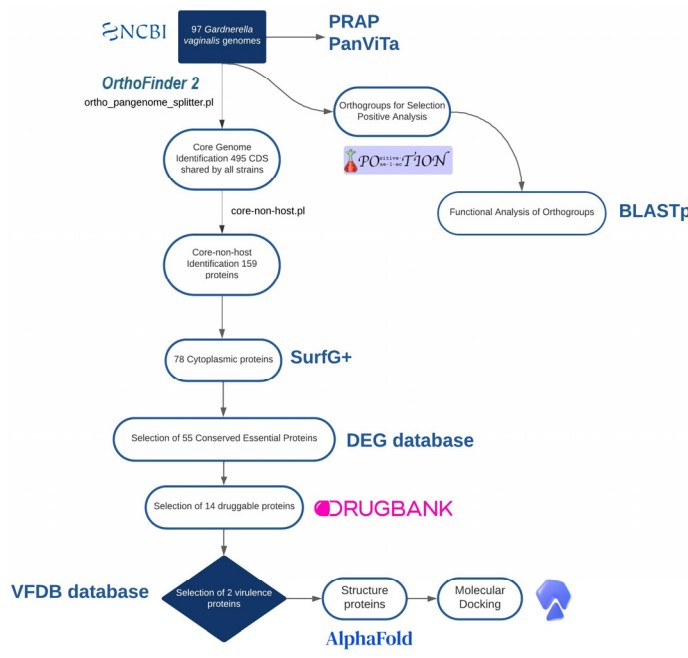

**Figure 1.** Workflow of methodologies used to select the drug candidates and genes related to Darwinian positive selection. Note: In this figure, a workflow with all the materials and methods used in this work can be observed, with the methods inside each of the oval balloons and the program symbol next to it.

### 3.1. Analysis of Positively Selected Orthogroups

Orthofinder selected 2568 orthogroups to predict positive Darwinian selection by Potion. After filtering, 231 groups were selected as valid. A total of 29 groups were classified in Model 8, chosen for having the best natural log-likelihood value (lnL value) of the observed data sequence given the model parameters. A cut-off point of 5% significance was used for the program's statistical analyses.

To perform a functional analysis of the orthogroups that were positively selected (Supplementary Table S2), we performed blast analyses (https://blast.ncbi.nlm.nih.gov/Blast.cgi) (accessed on 20 November 2023) with the blastp option. Of the 29 groups, 19 were hypothetical groups with no known functions. Among the other 10 groups, several are involved in ion, amino acid, and carbohydrate transport processes and in cell wall biogenesis and transcriptional movements. In addition, one group is engaged with antibiotic efflux, for example, gene *Mef(A)*, which confers resistance to macrolides.

### 3.2. Resistance and Virulence Analysis

For pan-resistome analysis (Figure 2A), no genes belonging to the core resistome were found, i.e., those found in all strains. However, the power law model was employed to perform the pan-resistome, generating a pan-resistome size $p = 2.078 * x^{0.197}$ ($R^2 = 0.803$). This R-value of less than 1 suggests an open pan-resistome, revealing that *G. vaginalis* can acquire resistance genes from other organisms throughout its evolution.

Through the CARD, the PRAP software v1.0 application also found gene *IsaC* in two lineages that provides resistance to lincosamide, pleuromutilin, and streptogramin antibiotic. The gene 'mel' was found in seventeen lineages, possibly providing resistance to macrolide and streptogramin antibiotics. The gene *mef(A)* was found in thirteen, providing resistance to macrolide antibiotic. Finally, the genes *tet(M)* and *tet(L)* were found in nineteen and two lineages, respectively, and both provide resistance against tetracycline antibiotics (Figure 2B).

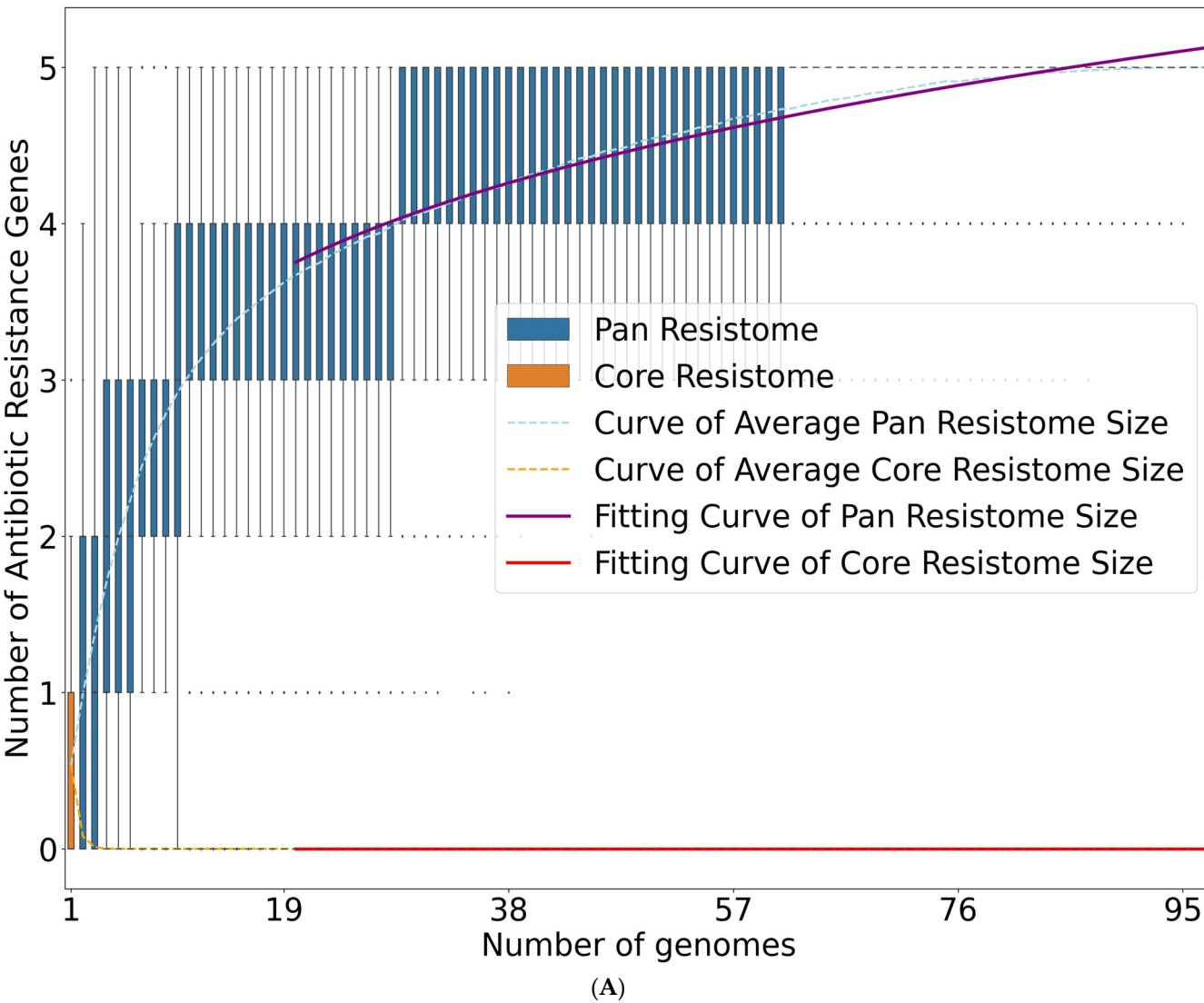

**Figure 2.** *Cont.*

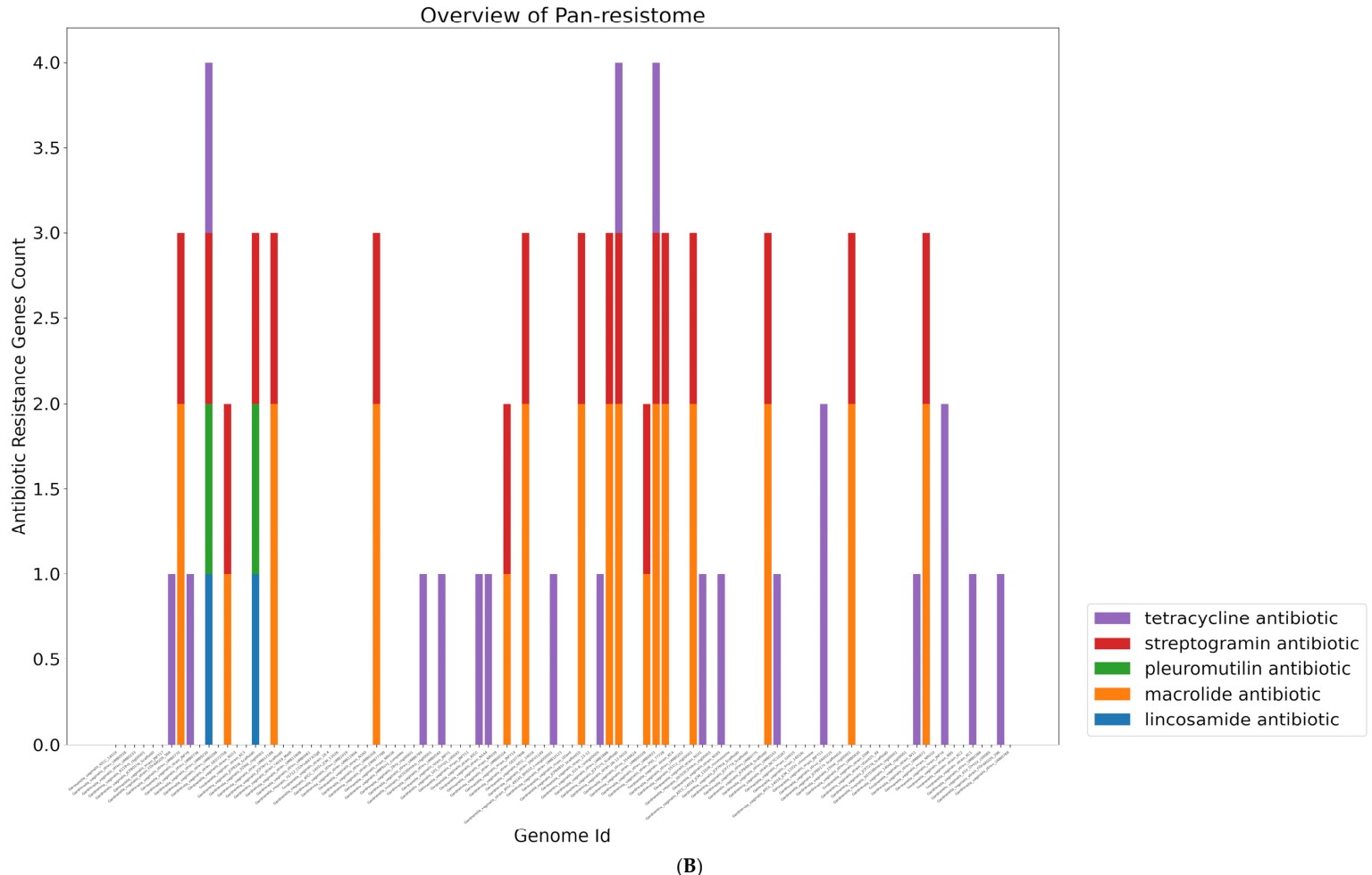

**Figure 2.** Analysis of pan-resistome and resistance genes using PRAP software v1.0. Note: (**A**) shows a growth curve of pan-resistome (in blue) and core resistome (in orange). In addition, dotted lines represent a curve of average and the fitting curve for pan-resistome and core resistome. In (**B**) the number of antibiotic resistance genes for each antibiotic class in different colors can be observed, such as tetracycline in purple, streptogramin in red, pleuromutilin in green, macrolide in orange, and lincosamide in blue.

Using PanViTa, we found *sigA/rpoV* and *msbB* alleles related to virulence, which were predicted by using the VFDB (Supplementary Table S3).

### 3.3. Selection of Drug Targets

Initially, proteins were selected based on the orthogroup they belonged to, using Orthofinder and *in-house* scripts (ortho_pangenome_splitter.pl). The analysis encompassed all proteins identified in the core, resulting in 495 proteins across all genomes. From this pool, only 159 proteins were chosen (core-non-host.pl) due to their non-homology with the host, a precaution taken to mitigate potential adverse effects. In the search for proteins with cytoplasmic localization for use as drug targets, SurfG+ identified and selected only 78 proteins.

The remaining proteins underwent scrutiny based on essentiality (n = 55) for the species, druggability (n = 14), and virulence, employing DEG, VFDB, and DrugBank programs, respectively. Ultimately, two proteins, WP_004132099.1 (RNA polymerase sigma factor sigA) (gene *sigA*) and WP_004131683.1 (UDP-N-acetylenolpyruvoylglucosamine reductase) (gene *murB*), were chosen as potential drug targets for subsequent structural and molecular docking analyses.

### 3.4. Prediction of the Tertiary Structure of the Selected Proteins

Alphafold was used to predict the tertiary structure of two previously selected proteins, WP_004132099.1 and WP_004131683.1 (Figure 3). For each, 24 models were predicted after reordering by model confidence, containing the prediction with the (i + 1)-th highest confidence, predicted from the pLDDT in which "ranked_0.pdb" has the highest confidence.

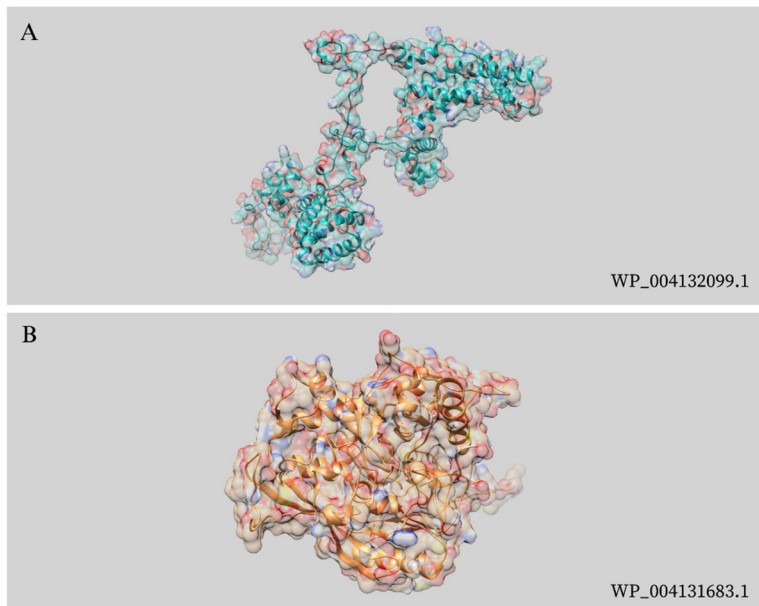

**Figure 3.** Tertiary structure of WP_004132099.1 and WP_004131683.1 predict by Alphafold. Note: Image (**A**) shows WP_004132099.1 tertiary structure predicts by Alphafold (in blue). Image (**B**) shows WP_004131683.1 tertiary structure predicts by Alphafold (in orange).

The predicted Ramachandran diagrams showed a high quality of the structure of the predicted proteins and are present in the Supplementary Material (Figures S1 and S2).

### 3.5. Molecular Docking of Selected Proteins with Natural Compounds

The previously selected proteins WP_004132099.1 and WP_004131683.1 were used for molecular docking analysis with 5008 natural compounds. The best ligands were, respectively, DLNC_ZINC08635277 with a free energy score of -9.68 and a hydrogen bond on Tyrosine 250 belonging to the active site of protein WP_004132099.1 and DLNC_ZINC03840479

with a score of –9.689 with three hydrogen bonds on residues Lysine 166, Phenylalanine 202, and Leucine 51 of the active site of protein WP_004131683.1 (Figure 4 and Table 2). This interactions in 2D are illustrated in Supplementary Material (Figures S3 and S4).

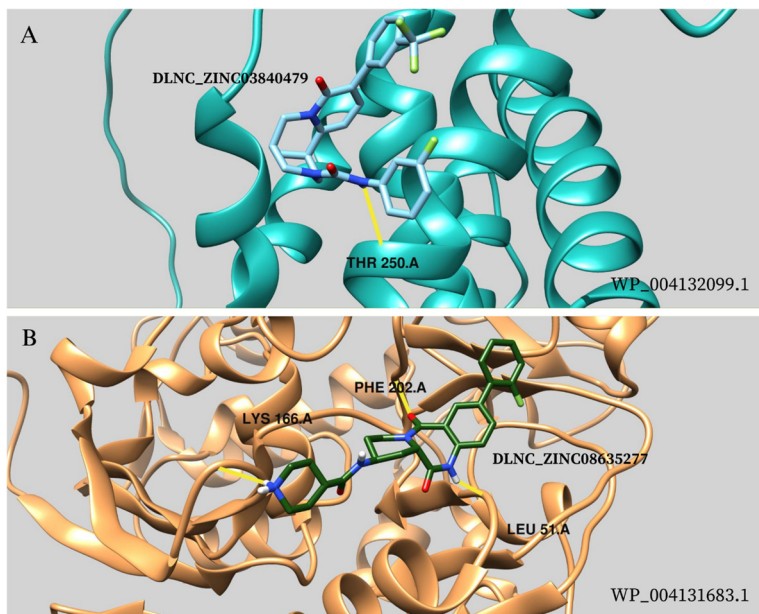

**Figure 4.** Docking molecular analyses of the proposed new drug targets. Note: In (**A**), the protein WP_004132099.1 can be observed in blue color and its best compound ligand DLNC_ZINC08635277 in shades of blue color. The link between the protein and its ligand by the hydrogen bridge with the amino acid Tyrosine 250 (THR 250) is stained yellow. In (**B**), the protein WP_004131683.1 in orange color and its best compound ligand DLNC_ZINC03840479 in shades of green color can be observed. The link between the protein and its ligand via the hydrogen bridge with the amino acids Lysine 166 (LYS 166), Phenylalanine 202 (PHE 202), and Leucine 51(LEU 51) is stained yellow.

**Table 2.** Proteins selected for drug targets with their predicted natural compounds from the ZINC database, hydrogen bonding, and residue analyzed by Chimera software.

| Protein | ZINC Compounds | Auto Dock Vina | Hbonds Number | Residue | Ångström (Å) |
|---|---|---|---|---|---|
| WP_004132099.1 | DLNC_ZINC08635277 | −9.68 | 1 | THR 250 | 2.291 Å |
| WP_004131683.1 | DLNC_ZINC03840479 | −9.689 | 3 | LYS 166 | 2.123 Å |
| | | | | PHE 202 | 2.327 Å |
| | | | | LEU 51 | 2.227 Å |

## 4. Discussion

*G. vaginalis* is a species of great medical importance, especially for women's health. Recent studies have shown its great gene diversity, antibiotic resistance, and virulence phenotype [49,50]. Comparative analyses were carried out to evaluate the positive Darwinian selection of all the annotated genomes deposited in the NCBI database of *G. vaginalis*.

Positive Darwinian selection is understood as characteristics that have been selected over time to maintain phenotypes that are considered advantageous for the survival of the species. What occurs are mutations that lead to amino acid changes in the protein that are targets of natural selection, i.e., non-synonymous substitution [51]. Antibiotic resistance may be one of these mechanisms that are evolutionarily selected. Through phylogenetic comparison of protein-coding genes, Darwinian positive selection analyses showed relevant orthogroups that were selected, which shows that *G. vaginalis* is a species with high genetic diversity and that genes have been added through evolution [52]. Fundamental groups that were analyzed are the *Mef(A)* gene, which is related to resistance

to macrolide class antibiotics [53], and *tet(M)* and *tet(L)*, which are related to tetracycline resistance [54]. This relationship shows how much this pathogenic species has evolved towards the resistance phenotype.

In addition, biofilm formation is an important characteristic of *Gardnerella vaginalis* and others various bacterial that favors horizontal gene transfer, contributing to pathogenicity and antimicrobial resistance, and is an important mechanism in the evolution of organisms and the selection of beneficial traits [55,56]. In addition, studies have shown a high genomic plasticity of *G. vaginalis* strains, demonstrating that HGT can occur, suggesting a high capacity to acquire genes that may be related to antibiotic resistance [22–24]. The interaction between *G. vaginalis* and species that can cause STIs is enhanced by the formation of biofilms by this pathogen, which it serves to stabilise a relationship of synergism and to enhanceresistance to antibiotics [56].

The resistance and virulence of these strains were then predicted to correlate with the orthogroup positively selected by PRAP and PanVita, respectively. These analyses showed resistance to the macrolide group class, which is one of the main choices for the treatment of sexually transmitted infections such as chlamydia and gonorrhea [3,56]. Although *G. vaginalis* is a dysbiosis in the vaginal microbiota, it increases the risk of acquiring sexually transmitted infections, mainly related to the formation of biofilms, which can contribute to the acquisition of genes by horizontal transfer, which can be resistance genes. In addition, this virulence factor may be associated with the upregulation of genes that confer antimicrobial resistance [6–8]. Another comparative genomics study demonstrated the presence of two genes related to resistance to this class of antibiotic as well as one major facilitator superfamily (MFS) transporter, as well as four unknown multidrug efflux systems [25].

In addition, when analyzing the pan-resistome using the power law model, the R-value was lower than 1, suggesting an open pan-resistome; this can be understood as a greater capacity to acquire resistance genes, as shown by the analysis of which genes were positively selected. Furthermore, many singletons show the relationship of resistance genes that evolve with the species, guaranteeing this phenotype for *G. vaginalis* [57].

In light of the aforementioned result and the potential for *G. vaginalis* to persist in acquiring resistance genes throughout its evolutionary trajectory, it is plausible that it has already developed such genes. Consequently, reverse vaccinology and molecular docking analyses were conducted.. Firstly, within subtractive genomics, the genes present in all the strains, i.e., those that belonged to the core genome, were compared in terms of their presence in their host to avoid possible adverse reactions. The proteins encoded by these genes, which were only present in the bacteria, were tested for their subcellular localization. Proteins considered to be cytoplasmic are usually used as drug targets because they are more likely to be involved in the microorganism's survival mechanisms [58], so 78 proteins went on to be analyzed for essentiality.

After the essentiality analysis, these proteins were tested against databases to assess whether they could bind to antibiotic compounds and were already considered virulence mechanisms among the most diverse organisms. Ultimately, two proteins were deemed to meet all quality criteria and underwent a molecular docking analysis to ascertain their potential for binding with naturally occurring compounds that may possess antibiotic properties.The first protein, WP_004132099.1, is a RNA polymerase sigma factor sigA. This protein has the vital function of directing the RNA polymerase enzyme to its promoter, which allows the bacterial transcription process to begin [59,60]. Drugs that involve this protein as a drug target are already widely studied, for example, acting as an inhibitor of infection caused by *Staphylococcus aureus* [61] or even tuberculosis [62,63]. In addition, this gene is related to the putative transcriptional regulator *WhiB7*, which is crucial for resistance to various classes of antibiotics; this has been demonstrated in *Mycobacterium smegmatis,* and so, inhibiting this WhiB7 binding could make ineffective antibiotics effective again [64]. Finally, a detail that is striking about *G. vaginalis* is that some species present the *sigA (RpoV)* gene as a virulence factor, predicted by the PanVita software v1.0 application

in five strains, also called sigma A, and involved in this transcription initiation process. For these reasons, this drug target is essential, especially if it is to be used against the diseases caused by these bacteria.

The second protein, WP_004131683.1, is an UDP-N-acetylenolpyruvoylglucosamine reductase. This protein catalyzes the reaction of UDP-N-acetylmuramic acid, an essential part of peptidoglycan. Peptidoglycan is responsible for maintaining the structure of the cell wall of prokaryotes [65,66]. Because it is a molecule that plays an essential role in forming the cell wall and maintaining the structure and, consequently, the presence of the microorganism in the host, it is often used in studies as a drug target. In silico work using *Acinetobacter baumanii*, *Escherichia coli*, and *Pseudomonas aeruginosa*, and even in vitro studies against *Corynebacterium glutamicum*, demonstrated that the *murB* gene, which encodes this protein, had important effects on maintaining the survival of microorganisms and could be used as drug targets [66–68].

## 5. Conclusions

In conclusion, the *G. vaginalis* genomes showed the selection of several positively selected genes, especially those related to resistance and virulence. In addition, the species' pan-resistome is open, demonstrating that in addition to the genes already found, it may be able to acquire new genes throughout evolution. This shows that this phenotype is essential for maintaining the species in its pathogenic trait during its evolution.

Coupled with this, two proteins with the potential to promote virulence were predicted, selected by subtractive genomics. Given the resistance observed, molecular docking analyses were carried out on these proteins with natural compounds to choose them in silico as potential drug targets. These proteins were linked with two natural compounds with good binding energy; in addition to that, these proteins reveal themselves to be excellent targets due to their functions in critical survival processes because they are also linked to the resistance and virulence genes found here and have already been tested against other microorganisms. These targets should also be tested in vitro and in vivo, coupled with techniques to evaluate the biofilm-forming capacity of these bacteria, to find more promising results against *G. vaginalis*.

**Supplementary Materials:** The following supporting information can be downloaded at https://www.mdpi.com/article/10.3390/venereology3030010/s1, Table S1: Total number of positively selected orthogroups, where "P" for groups with q values less than the cut-off point of 0.05, "u" for groups with q values greater than the cut-off point and p values less than the cut-off "n" for negative groups; Table S2: Result functional analysis of all of orthologous groups positive selected by Potion software; Table S3: Alignment between the 97 genomes of Gardnerella vaginalis with virulence from the VFDB database predicted by PanViTa; Figure S1: Ramachandran plot predicted by PROCHECK of protein WP_004131683.1; Figure S2: Ramachandran plot predicted by PROCHECK of protein WP_004132099.1; Figure S3: 2D structure between the interaction of DLNC_ZINC08635277 with protein WP_004132099.1; Figure S4: 2D structure between the interaction of DLNC_ZINC03840479 with protein WP_004131683.1.

**Author Contributions:** A.G.F. and E.G.S. performed the download, all data processing and analysis, participated in the study design, and wrote this manuscript; F.V.D. helped in the data processing and in the writing of this manuscript; A.K.J., M.L.C.P., L.P.R. and L.G.R.G. helped in the data processing; V.A.d.C.A. coordinated the study; S.d.C.S. designed, coordinated the study, helped with analysis, and wrote this manuscript. All authors have read and agreed to the published version of the manuscript.

**Funding:** This research received no external funding.

**Institutional Review Board Statement:** Not applicable.

**Informed Consent Statement:** Not applicable.

**Data Availability Statement:** Genome sequences are available in the NCBI database.

**Acknowledgments:** We thank the agencies of the Brazilian Federal Agency for Support and Evaluation of Graduate Education (CAPES) and the Minas Gerais Research Funding Foundation (FAPEMIG).

We give our thanks to the post-graduate program in Genetics at the Federal University of Minas Gerais and the post-graduate program in Medicina Tropical e Infectologia at the Federal University of Triângulo Mineiro.

**Conflicts of Interest:** The authors declare no conflicts of interest.

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
