# Peer review of "Unveiling Resistance and Virulence Mechanisms under Darwinian Positive Selection for Novel Drug Discovery for Gardnerella vaginalis"

_venereology, doi:10.3390/venereology3030010_

Round 1

Reviewer 1 Report

Comments and Suggestions for Authors

The authors have dealt with an interesting and valuable topic, given the many discussions in terms of pathogenicity and clinical significance for women. In the work, modern techniques were used for analysis, which increases the value of the presented research. However, what raises questions in this work is the scope of the antibiotics analyzed. The authors repeatedly refer to the treatment of BV and infections with macrolide antibiotics, indicating them as first-line drugs. They emphasize their importance, growing resistance, and they are also included in the results and charts. However, these are not first-line drugs for such infections, it is, for example, metronidazole. The authors also refer repeatedly to a publication (Schuyler, J.A.; Chadwick, S.G.; Mordechai, E.; Adelson, M.E.; Gygax, S.E.; Hilbert, D.W. Draft Genome Sequence of a Metronidazole-Resistant Gardnerella Vaginalis Isolate. Genome Announc. 2015, 3, doi:10.1128/GENOMEA.00992-15.), which also highlights the importance of metronidazole resistance rather than macrolide resistance. Hence, it is the macrolide analysis, which needs to be verified, that raises doubts, in the context of the Mef(A) genes analyzed.

Comments on the Quality of English Language

No comments.

Author Response

Comments 1: The authors have dealt with an interesting and valuable topic, given the many discussions in terms of pathogenicity and clinical significance for women. In the work, modern techniques were used for analysis, which increases the value of the presented research. However, what raises questions in this work is the scope of the antibiotics analyzed. The authors repeatedly refer to the treatment of BV and infections with macrolide antibiotics, indicating them as first-line drugs. They emphasize their importance, growing resistance, and they are also included in the results and charts. However, these are not first-line drugs for such infections, it is, for example, metronidazole. The authors also refer repeatedly to a publication (Schuyler, J.A.; Chadwick, S.G.; Mordechai, E.; Adelson, M.E.; Gygax, S.E.; Hilbert, D.W. Draft Genome Sequence of a Metronidazole-Resistant Gardnerella Vaginalis Isolate. Genome Announc. 2015, 3, doi:10.1128/GENOMEA.00992-15.), which also highlights the importance of metronidazole resistance rather than macrolide resistance. Hence, it is the macrolide analysis, which needs to be verified, that raises doubts, in the context of the Mef(A) genes analyzed.

Response 1: Thank you for pointing this out. We agree with this comment. Therefore, we have made some changes to this topic in our work, clarifying it more. Where we pointed out that macrolides are not really the first line of treatment for Gardnerella vaginalis, but metranizadol is. However, infections caused by Gardnerella vaginalis are associated with an increased risk of contracting sexually transmitted infections. For example, BV can increase a woman's risk of contracting Neisseria gonorrhoeae, Chlamydia trachomatis and Trichomonas, which can be treated with macrolides, as in the case of chlamydia and gonorrhea. Macrolides are used to treat various Gram-positive bacteria. Joint treatment can be a problem when it comes to antibiotic resistance in these microorganisms (https://doi.org/10.3389/fmicb.2016.00747).

Reviewer 2 Report

Comments and Suggestions for Authors

Comments to the authors:

The authors present a manuscript related to the analysis of 29 orthologous groups of gene sequences of the Gardnerella vaginal pathogen through bio-informatic analysis suggesting that the G. vaginalis genome presents two proteins by in silico analysis that may be potential drug targets. Therapeutics but it is necessary to develop in vitro and in vivo studies in the future to evaluate the possibilities of functionality of these proteins

It is important that authors review certain points in their manuscript

1.-The authors must review the presentation of their manuscript since throughout the text the spaces are not justified line # 45, #90, they are missing some punctuation # 98,

2.-The same previous point is observed again on lines # 106 and so on

3.- In figure 1, 2 and 3

I suggest increasing the resolution, it is not clearly observed and they are important aspects of the results section

Author Response

Comments 1: The authors present a manuscript related to the analysis of 29 orthologous groups of gene sequences of the Gardnerella vaginal pathogen through bio-informatic analysis suggesting that the G. vaginalis genome presents two proteins by in silico analysis that may be potential drug targets. Therapeutics but it is necessary to develop in vitro and in vivo studies in the future to evaluate the possibilities of functionality of these proteins.

Response 1: Thank you for pointing this out. We agree with this comment, our study focused on the analysis of 29 orthologous groups of gene sequences from the pathogen Gardnerella vaginalis with the aim of better understanding the bacterium's phenotype in the context of resistance and virulence. Through bioinformatic analysis, we have identified two proteins which, from our in-silico study, show potential as therapeutic targets.

We would like to emphasize that the in-silico approach we used was chosen as an initial virtual screening step, allowing the preliminary identification of potential drug targets in an efficient and cost-effective manner. This methodology is widely used for screening large genomic datasets and predicting protein-drug interactions before moving on to more costly and time-consuming in vitro and in vivo studies.

We recognize the importance of additional experimental validations and agree that in vitro and in vivo studies are essential to assess the functionality and therapeutic efficacy of the identified proteins. As mentioned in the manuscript, our work proposes these proteins as potential initial targets, and future experimental work will be necessary to confirm these findings and explore their therapeutic applications.

Comments 2: It is important that authors review certain points in their manuscript

1.-The authors must review the presentation of their manuscript since throughout the text the spaces are not justified line # 45, #90, they are missing some punctuation # 98,

2.-The same previous point is observed again on lines # 106 and so on

Response 2: Agree. Regarding the justified text, we have revised the formatting and corrected it.

Comments 3: In figure 1, 2 and 3

I suggest increasing the resolution, it is not clearly observed and they are important aspects of the results section

Response 3: The text submitted with the figures is in low resolution, but when we submit it to the journal, we send the images separately, I believe this is due to the journal's platform for submitting articles. However, we accept your suggestion and have increased the resolution of our images in the place indicated in the journal for submitting images.

Reviewer 3 Report

Comments and Suggestions for Authors

This is a study with a good idea; however, the reviewer believes that the manuscript needs to be improved.

Apparently, the manuscript should be subjected to language and scientific editing, since some of its fragments are difficult to understand. A typical example is in lines 61-69, where the formation of biofilms is explained by the presence of sialidase, and the biofilms themselves are attributed functions that they do not possess. Perhaps the authors should consult with experts in this field of research.

The title of the manuscript is “Unveiling Resistance and Virulence Mechanisms Under Darwinian Positive Selection for Novel Drug Discovery for Gardnerella vaginalis.” It could be assumed that the design of the manuscript involves the study of traits that appear as a result of natural selection in Gardnerella during the process of evolution. This is wrong. Overall, the manuscript describes existing features of Gardnerella genomes; the reviewer believes that manuscripts with such a title should use groups of strains that originate from the same source. Gardnerella does not appear to be suitable for this role because the strains used in this study were isolated from different sources and had in common only the fact that they belonged to the same species. How can we track traits that ensure positive selection in organisms not related by common ancestry? The design of this study does not involve studying the role of such characters in the evolution of Gardnerella, but merely states their presence. It would be understandable if there was evidence that Gardnerella can be transmitted, for example, as pathogens of nosocomial infections or pathogens of STIs; in this case, it is possible to track the evolution of resistance mechanisms, and so on.

However, the reviewer reiterates that the overall idea of this study is good. This comment particularly concerns the authors' idea to use data on the evolution of resistance and virulence mechanisms in order to create new drugs. But Gardnerella are hardly suitable here either. For example, based on Figure 2, antibiotic resistance genes were found in only 34 of the 97 genomes of G. vaginalis strains. Are these strains somehow related in origin? Don't clear.

In the “Introduction” section, it is necessary to indicate why the authors took Gardnerella as the object of study. Obviously, Gardnerella are not highly pathogenic bacteria; they are not as dangerous, for example, as the causative agents of nosocomial infections or the same STIs.

Author Response

Comments 1: Apparently, the manuscript should be subjected to language and scientific editing, since some of its fragments are difficult to understand. A typical example is in lines 61-69, where the formation of biofilms is explained by the presence of sialidase, and the biofilms themselves are attributed functions that they do not possess. Perhaps the authors should consult with experts in this field of research.

Response 1: Thank you for pointing this out. We agree with this comment. Therefore, we have made some changes to the formation of biofilms by the bacteria, which is a virulence factor, and the production of prolidases and sialidases, which are different factors. We have improved the wording to make the pathogenic potential of the bacteria clearer. (Page 2, lines 63 - 76).

Comments 2: The title of the manuscript is “Unveiling Resistance and Virulence Mechanisms Under Darwinian Positive Selection for Novel Drug Discovery for Gardnerella vaginalis.” It could be assumed that the design of the manuscript involves the study of traits that appear as a result of natural selection in Gardnerella during the process of evolution. This is wrong. Overall, the manuscript describes existing features of Gardnerella genomes; the reviewer believes that manuscripts with such a title should use groups of strains that originate from the same source. Gardnerella does not appear to be suitable for this role because the strains used in this study were isolated from different sources and had in common only the fact that they belonged to the same species. How can we track traits that ensure positive selection in organisms not related by common ancestry? The design of this study does not involve studying the role of such characters in the evolution of Gardnerella, but merely states their presence. It would be understandable if there was evidence that Gardnerella can be transmitted, for example, as pathogens of nosocomial infections or pathogens of STIs; in this case, it is possible to track the evolution of resistance mechanisms, and so on. However, the reviewer reiterates that the overall idea of this study is good. This comment particularly concerns the authors' idea to use data on the evolution of resistance and virulence mechanisms in order to create new drugs. But Gardnerella are hardly suitable here either. For example, based on Figure 2, antibiotic resistance genes were found in only 34 of the 97 genomes of G. vaginalis strains. Are these strains somehow related in origin? Don't clear.

Response 2: We have already carried out a phylogenetic analysis showing the common ancestry of the organisms studied. This analysis is fundamental to understanding the evolutionary relationships between the different Gardnerella vaginalis isolates. Although the isolates were obtained from different sources, the phylogenetic analysis confirms that they belong to the same ancestral group, allowing the identification of the resistance and virulence mechanisms that were the subject of our study. We emphasize that the diversity of sources does not affect the validity of the phylogenetic analysis (https://doi.org/10.3390/venereology2040012). This work has also shown that Gardnerella vaginalis has high genomic plasticity, which is crucial for adaptation and survival in different environments. This characteristic allows the presence of different resistance and virulence mechanisms, regardless of the origin of the isolates. Genomic plasticity is an important factor in the evolution of organisms and in the selection of beneficial traits. Regarding the title of the manuscript, we believe that it adequately reflects the aim of the study, which is to investigate mechanisms of resistance and virulence under Darwinian positive selection. Therefore, we do not believe it is necessary to change the title. We thank you again for your consideration and hope that our clarifications are satisfactory.

Comments 3: In the “Introduction” section, it is necessary to indicate why the authors took Gardnerella as the object of study. Obviously, Gardnerella are not highly pathogenic bacteria; they are not as dangerous, for example, as the causative agents of nosocomial infections or the same STIs.

Response 3: Although Gardnerella vaginalis is not considered a highly pathogenic bacterium like those that cause nosocomial infections or other STIs, it is the major microorganism associated with bacterial vaginosis (BV). BV is a common condition affecting many women and is associated with a high relapse rate after treatment. BV can contribute significantly to the development of other sexually transmitted infections. Has several virulence factors and resistance mechanisms that justify its importance as a research target. For example, biofilm formation is a key factor contributing to resistance to antimicrobial treatment, making it difficult to eradicate the infection and favoring the persistence and recurrence of BV. BV and Gardnerella vaginalis are often under-reported and neglected in clinical practice, despite their significant impact on women's reproductive health, have sometimes been associated with premature birth or miscarriages.

Thank you very much for your valuable comments. In response to your observations, we have improved the introduction and discussion of the manuscript, considering the points you highlighted. These improvements have been made with the aim of making our analysis clearer and more understandable.

Round 2

Reviewer 1 Report

Comments and Suggestions for Authors

It can be seen that the authors have tried to clarify some issues related to antibiotic therapy of BV involving G.vaginalis. However, the contents are still inconsistent and lead the reader to believe that macrolides are used to treat G.vaginalis and BV infections and the emergence of resistance to such a drug. Sample excerpt from lines 96-99. BV is not classified as an STI (STD) hence does not translate into an association of the use of this group of antibiotics in STI infections, despite the fact that the effect of G.vaginalis on predisposition to such infections is suggested. The authors in the title of their paper suggest the search for new drugs for G.vaginalis infections, and it is known that macrolides are not used in this case. The consideration of resistance at the genetic level is understandable. However, despite the amendments, the persistent author’s focus on the macrolide group usage in the STIs treatment caused by BV and G.vaginalis is hard to understand. For clinicians, the parameters for diagnosing BV are defined, as is the use of appropriate therapy. Consequently, for the viewer of the presented paper, the association of macrolide therapy for STIs with BV treatment failure remains difficult to grasp. It would be worthwhile to rewrite the paper more carefully to make it more coherent and logical.  Perhaps the authors had in mind an explanation of the mechanism of resistance to this group of antibiotics in G.vaginalis? However, it is difficult to unequivocally draw such a conclusion from this paper, also based on excerpt 366-371. Hence, it is difficult to understand the authors' message, despite the valuable research conducted.

Comments on the Quality of English Language

No comments

Author Response

Comments 1: It can be seen that the authors have tried to clarify some issues related to antibiotic therapy of BV involving G.vaginalis. However, the contents are still inconsistent and lead the reader to believe that macrolides are used to treat G.vaginalis and BV infections and the emergence of resistance to such a drug. Sample excerpt from lines 96-99. BV is not classified as an STI (STD) hence does not translate into an association of the use of this group of antibiotics in STI infections, despite the fact that the effect of G.vaginalis on predisposition to such infections is suggested. The authors in the title of their paper suggest the search for new drugs for G.vaginalis infections, and it is known that macrolides are not used in this case. The consideration of resistance at the genetic level is understandable. However, despite the amendments, the persistent author’s focus on the macrolide group usage in the STIs treatment caused by BV and G.vaginalis is hard to understand. For clinicians, the parameters for diagnosing BV are defined, as is the use of appropriate therapy. Consequently, for the viewer of the presented paper, the association of macrolide therapy for STIs with BV treatment failure remains difficult to grasp. It would be worthwhile to rewrite the paper more carefully to make it more coherent and logical.  Perhaps the authors had in mind an explanation of the mechanism of resistance to this group of antibiotics in G.vaginalis? However, it is difficult to unequivocally draw such a conclusion from this paper, also based on excerpt 366-371. Hence, it is difficult to understand the authors' message, despite the valuable research conducted.

Response 1: Thank you for pointing this out. We agree with this comment. Therefore, we have improved our understanding that macrolide therapy should not be used to treat bacterial vaginosis, we've removed and rewritten the introduction for better understanding. We have made it clearer that resistance to some antibiotics is a recurring problem in several studies, including this one. In addition, we have proposed new drugs in view of this high level of resistance to several groups of antibiotics, as well as the high genomic plasticity and the presence of genomic islands in these Gardnerella vaginalis genomes (https://doi.org/10.1128/jb.00056-12; https://doi.org/10.1186/1471-2180-12-301; https://doi.org/10.3390/venereology2040012). The proposal of new targets is related to potential proteins as drug targets, through a virtual screening where two targets were proposed that are inhibited by natural products. Since this bacterium has a high genomic plasticity and therefore the ability to acquire genes from other microorganisms, which suggests the possibility of acquiring resistance genes, since a previous study showed the presence of resistance islands, as well as the phenotype of resistance and virulence observed throughout the evolution of this microorganism.

In addition, we have added more references throughout the text for a better discussion of this work: https://doi.org/10.4248/IJOS11026; https://doi.org/10.1038/nrmicro821; https://doi.org/10.1128/jb.00056-12; https://doi.org/10.1093/femsre/fuz027.

Line 68 – 72 we discussed more about the biofilms that are related to this bacterium, as well as the relationship between this and the upregulation of genes enabling antimicrobial resistance.

In the discussion of the work and conclusion we have rewritten and given greater relevance to our results, line 356-359 and line 428.

Reviewer 3 Report

Comments and Suggestions for Authors

The authors have done significant work that has improved the quality of the manuscript; however, the reviewer believes that changes to the manuscript need to be made that are clearly necessary.

In his previous review, the reviewer mentioned the need for linguistic and scientific editing of the manuscript since some of its fragments are difficult to understand. The reviewer strongly recommends that the authors seek advice from specialists in microbiology. This remark is related to the interpretation of the results in the “Discussion” section.

For example, in lines 367-372, the authors conclude that the presence of macrolide resistance genes in Gardnerella may be the reason for the ineffectiveness of STI treatment. At the same time, the authors base their conclusion on studies that do not talk about this. So, in [3] (Shaskolskiy, B.; Dementieva, E.; Leinsoo, A.; Runina, A.; Vorobyev, D.; Plakhova, X.; Kubanov, A.; Deryabin, D.; Gryadunov, D . Drug Resistance Mechanisms in Bacteria Causing Sexually Transmitted Diseases and Associated with Vaginosis Front. 2016, 7.) talks about the mechanisms that lead to resistance to antimicrobial drugs and the like; however, this study does not provide evidence to support the authors' claims. In [54] (Bostwick, D.G.; Woody, J.; Hunt, C.; Budd, W. Antimicrobial Resistance Genes and Modeling of Treatment Failure in Bacterial Vaginosis: Clinical Study of 289 Symptomatic Women. J. Med. Microbiol. 2016, 65, 377–386) indicate a wide distribution of antibiotic resistance genes in the vaginal microbiome in women with bacterial vaginosis. [15] describes the genome sequence of a vaginal isolate of Gardnerella vaginalis that is highly resistant to metronidazole. The reviewer would like to note that to confirm the authors’ conclusions in this case, data on the possibility of transfer of antibiotic resistance genes from Gardnerella to other bacteria, for example, pathogens of STIs, would be more suitable. The reviewer does not deny the presence of well-documented difficulties in the treatment of STIs occurring against the background of BV, but, nevertheless, calls on the authors to use the data of other researchers more correctly. Is it possible to assume that the biofilms formed by Gardnerella allow STI pathogens to hide? And is it possible to transmit antibiotic resistance genes in these biofilms? And so on.

At the same time, the reviewer would like to note that the section clearly lacks an interpretation of the results obtained and an explanation of the authors’ conclusions.

Author Response

Comments 1: The authors have done significant work that has improved the quality of the manuscript; however, the reviewer believes that changes to the manuscript need to be made that are clearly necessary.

In his previous review, the reviewer mentioned the need for linguistic and scientific editing of the manuscript since some of its fragments are difficult to understand. The reviewer strongly recommends that the authors seek advice from specialists in microbiology. This remark is related to the interpretation of the results in the “Discussion” section.

For example, in lines 367-372, the authors conclude that the presence of macrolide resistance genes in Gardnerella may be the reason for the ineffectiveness of STI treatment. At the same time, the authors base their conclusion on studies that do not talk about this. So, in [3] (Shaskolskiy, B.; Dementieva, E.; Leinsoo, A.; Runina, A.; Vorobyev, D.; Plakhova, X.; Kubanov, A.; Deryabin, D.; Gryadunov, D . Drug Resistance Mechanisms in Bacteria Causing Sexually Transmitted Diseases and Associated with Vaginosis Front. 2016, 7.) talks about the mechanisms that lead to resistance to antimicrobial drugs and the like; however, this study does not provide evidence to support the authors' claims. In [54] (Bostwick, D.G.; Woody, J.; Hunt, C.; Budd, W. Antimicrobial Resistance Genes and Modeling of Treatment Failure in Bacterial Vaginosis: Clinical Study of 289 Symptomatic Women. J. Med. Microbiol. 2016, 65, 377–386) indicate a wide distribution of antibiotic resistance genes in the vaginal microbiome in women with bacterial vaginosis. [15] describes the genome sequence of a vaginal isolate of Gardnerella vaginalis that is highly resistant to metronidazole. The reviewer would like to note that to confirm the authors’ conclusions in this case, data on the possibility of transfer of antibiotic resistance genes from Gardnerella to other bacteria, for example, pathogens of STIs, would be more suitable. The reviewer does not deny the presence of well-documented difficulties in the treatment of STIs occurring against the background of BV, but, nevertheless, calls on the authors to use the data of other researchers more correctly. Is it possible to assume that the biofilms formed by Gardnerella allow STI pathogens to hide? And is it possible to transmit antibiotic resistance genes in these biofilms? And so on. 

At the same time, the reviewer would like to note that the section clearly lacks an interpretation of the results obtained and an explanation of the authors’ conclusions.

Response 1: Thank you for pointing this out. We agree with this comment. Therefore, I/we have improved the description of our results as well as the writing of our entire paper. Specifically, in line 362-372, now line 354-371, we have emphasised the importance of biofilm as one of the main virulence factors of Gardnerella vaginalis, which suggests be related to the increased risk of sexually transmitted infections, and how it may also increase the horizontal transfer of genes as well as the upregulation of genes conferring antimicrobial resistance: “In addition, biofilm formation is an important characteristic of Gardnerella vaginalis and others various bacterial that favors horizontal gene transfer, contributing to pathogenicity and antimicrobial resistance, and is an important mechanism in the evolution of organisms and the selection of beneficial traits [56,57]. As well as studies have shown a high genomic plasticity of G. vaginalis strains, demonstrating that HGT can be occurs, suggesting a high capacity to acquire genes that may be related to antibiotic re-sistance [23–25]. The interaction between G. vaginalis and species that can cause STIs is enhanced by the formation of biofilms by this pathogen, which stabilizes a relationship of synergism, as well as increased resistance to antibiotics [57]. The resistance and virulence of these strains were then predicted to correlate with the orthogroup positively selected by PRAP and PanVita, respectively. These analyses showed resistance to the macrolide group class, which is one of the main choices for the treatment of sexually transmitted infections such as chlamydia and gonorrhea [3,57]. Although G. vaginalis is a dysbiosis in the vaginal microbiota, it increases the risk of acquiring sexually transmitted infections, mainly related to the formation of biofilms, which can contribute to the acquisition of genes by horizontal transfer, which can be resistance genes. In addition, this virulence factor may be associated with the up-regulation of genes that confer antimicrobial resistance [6–8].”.

Biofilm formation was discussed more fully in the introduction, line 64-72: “This bacillus exhibits a higher pathogenic potential compared to other organisms, some studies have already identified the potential of this species, which is often un-derestimated. That can be explained by its ability to form biofilms [4], which is im-portant for increasing its chances of survival in the human body [5]adhering more in-tensely to vaginal epithelial cells. The biofilm is a characteristic of various bacterial that can be impedes the drug diffusion to the core, phenotypic changes, bacterial communication through quorum-sensing, upregulation of genes enabling antimicrobi-al resistance, presence of enzymes hindering drug penetration into the biofilm, and adaptation in physiological structure [6–8].”.

In this way, we can better discuss what we found and when we compare it with the data found in other articles (https://doi.org/10.3390/venereology2040012) we can suggest that biofilms are very important for G. vaginalis. Unfortunately, through computer analyses, we can't identify and analyze the biofilm in more depth as if we had made a culture of these organisms that we used in the work, but this already demonstrates a great advance in understanding for the literature to be able to deepen its look/research into this great virulence factor, which may be one of those responsible for HGT). We also agree/are in favor of carrying out more in vitro and in vivo tests to identify this biofilm.